# Gastrodin Improves Cognitive Dysfunction in REM Sleep-Deprived Rats by Regulating TLR4/NF-κB and Wnt/β-Catenin Signaling Pathways

**DOI:** 10.3390/brainsci13020179

**Published:** 2023-01-21

**Authors:** Bo Liu, Fei Li, Yunyan Xu, Qin Wu, Jingshan Shi

**Affiliations:** Key Lab for Basic Pharmacology of Ministry of Education, Zunyi Medical University, Zunyi 563000, China

**Keywords:** Gastrodin (GAS), cognitive dysfunction, REM sleep-deprived rats, TLR4/NF-κB signaling pathways, Wnt/β-catenin signaling pathways

## Abstract

Gastrodin is the active ingredient in *Gastrodia elata.* Our previous studies demonstrated that gastrodin ameliorated cerebral ischemia–reperfusion and hypoperfusion injury and improved cognitive deficit in Alzheimer’s disease. This study aims to examine the effects of gastrodin on REM sleep deprivation in rats. Gastrodin (100 and 150 mg/kg) was orally administered for 7 consecutive days before REM sleep deprivation. Seventy-two hours later, pentobarbital-induced sleep tests and a Morris water maze were performed to measure REM sleep quality and learning and memory ability. Histopathology was observed with hematoxylin–eosin staining, and the expression of the NF-κB and Wnt/β-catenin signaling pathways was examined using Western blot. After REM sleep deprivation, sleep latency increased and sleep duration decreased, and the ability of learning and memory was impaired. Neurons in the hippocampal CA1 region and the cortex were damaged. Gastrodin treatment significantly improved REM sleep-deprivation-induced sleep disturbance, cognitive deficits and neuron damage in the hippocampus CA1 region and cerebral cortex. A mechanism analysis revealed that the NF-κB pathway was activated and the Wnt/β-catenin pathway was inhibited after REM sleep deprivation, and gastrodin ameliorated these aberrant changes. Gastrodin improves REM sleep-deprivation-induced sleep disturbance and cognitive dysfunction by regulating the TLR4/NF-κB and Wnt/β-catenin signaling pathways and can be considered a potential candidate for the treatment of REM sleep deprivation.

## 1. Introduction

Sleep is a universal need of all higher life forms including humans, and humans spend about a third of their time asleep. Sleep deprivation refers to a sleep-time depression that cannot meet physiological needs due to several factors which causes various functional disorders such as endocrine, metabolic, cardiovascular and nervous system diseases [1]. With the development of modern society and the quickening of the pace of life, sleep deprivation has become one of the most important problem affecting the work and life of human beings [2]. Non-rapid eye movement (non-REM) is followed by a significantly shorter period of rapid eye movement (REM). Sleep deprivation, in particular, REM sleep deprivation, can affect mood, learning and memory function and can even lead to the development of neuropsychiatric disorders such as Alzheimer’s disease [3,4,5], Parkinson’s disease [6], depression [7] and so on. 

An increasing number of studies have found that REM sleep deprivation occurred before a decline in cognitive function. A lack of REM sleep has been shown to produce deficits in memory consolidation and plays an important role in brain development and brain plasticity in the several developmental stages of the human brain [8]. Animals also experience learning and memory problems after REM sleep deprivation [9]. However, the mechanism is still inconclusive [10]. At present, sedative and hypnotic drugs, such as benzodiazepines [11], zolpidem [12] and selective serotonin reuptake inhibitors trazodone, agomelatine and mirtazapine [13,14], are widely used to improve sleep problems. Ultimately, these drugs improve sleep-disorder-induced learning and memory impairment. Sedative and hypnotic drugs have significant clinical efficacy, but long-term use can lead to drug dependence, withdrawal reactions and other side effects. Therefore, it has become an urgent problem to study the mechanism of REM sleep deprivation and develop safe and effective treatment methods.

*Gastrodia elata* Blume, as a dried root of orchids, has effects of sedation and hypnosis, invigorating Qi and tranquilizing the mind (Yiqi Anshen in Chinese) and relieving rheumatism and colds [15]. Modern studies have shown that the active ingredients of *Gastrodia elata* Blume can exert sleep-promoting effects [16]. The chemical components of *Gastrodia elata* Blume include phenols, glycosides and polysaccharides, among which gastrodin (GAS) has important biological activities [17]. GAS has various pharmacological activities in the central nervous system, such as anti-epilepsy, anti-Alzheimer’s disease, anti-Parkinson’s disease, etc. [18,19]. Our research group and others researchers have demonstrated that GAS is able to ameliorate cerebral ischemia–reperfusion injury [20,21] and chronic cerebral hypoperfusion injury [22,23,24]. Nevertheless, the effect of GAS on learning and memory impairment caused by REM sleep deprivation has not been studied and is the main goal of the present study.

The available evidence suggests that REM sleep deprivation causes learning and memory impairment due to neuroinflammation [25,26], and the toll-like receptor 4 (TLR4)/nuclear factor kappa B (NF-κB) pathway is involved in sleep-deprivation-induced neuroinflammation [11]. A variety of active ingredients in traditional Chinese medicine, such as Suan-Zao-Ren decoction [27] and curcumin [28], can reduce hippocampal inflammation by inhibiting the TLR4/NF-κB signaling pathway and then improving learning and memory impairment in sleep-deprived rats. GAS can also play a protective role by inhibiting the TLR4/NF-κB signaling pathway and reducing the expression of downstream inflammatory factors in Parkinson’s disease rat models [29]. However, whether GAS can ameliorate REM sleep-deprivation-induced cognitive impairment through the TLR4/NF-κB pathway remains unknown.

The Wnt/β-catenin signaling pathway comprises a family of proteins that play critical roles in cell proliferation and differentiation, and it plays an important role in the development, regeneration and degenerative diseases of the central nervous system. The deregulation of Wnt/β-catenin signaling often leads to various serious diseases, including diseases of the central nervous system [30]. Furthermore, the Wnt/β-catenin signaling pathway is involved in sleep-deprivation-induced hippocampal neuronal injury in mice, and adjusting the Wnt/β-catenin signaling pathway can improve neuron genesis and learning and memory ability in sleep-deprived mice. There are studies that show that GAS has inhibitory effects on activated microglia-mediated inflammation and cell proliferation by regulating the Wnt/β-catenin signaling pathway [31].

We hypothesized that GAS can improve learning and memory impairment in rats with REM sleep deprivation and that the mechanism is related to the regulation of the TLR4/NF-κB and Wnt/β-catenin signaling pathways. A Morris water maze test, pentobarbital sodium sleep latency test and hematoxylin–eosin (HE) staining were used to observe the effect of GAS on REM sleep-deprived rats, and Western blot analysis was used to explore the specific mechanisms to provide an experimental basis for the use of GAS in the treatment of REM sleep-related diseases.

## 2. Materials and Methods

### 2.1. Drugs

The molecular formula of gastrodin (CAS) is C_13_H_18_O_7_, the molecular weight is 286.28, and the purity is 98%. CAS was purchased from Puyi Biotechnology Co., Ltd. (Shanghai, China, Product no.: 62499-27-8). The other chemicals were reagent grade.

### 2.2. Animals

Specific-pathogen-free (SPF) grade adult male Sprague Dawley rats (weighing 260 ± 20 g) were purchased from Hunan STA laboratory animal CO., LTD (Chongqing, China, Certificate No. SCXK2021-0003). All animals were adaptively fed for a week before the experiment. The rats were maintained under a 12 h light/dark cycle (50% relative humidity, between 20 °C and 25 °C) with free access to feed and tap water. All animal experiments were approved by the Experimental Animal Ethics Committee of Zunyi Medical University (ZMU21-2201-167). Sixty Sprague Dawley rats were randomly divided into the control group, model group, low-dose gastrodin group (100 mg/kg) and high-dose gastrodin group (150 mg/kg), the dosage were consistent with prior publications [22,32]. All rats were given preventive administration of GAS or normal saline (NS) by gavage daily for 7 days before REM sleep deprivation. To avoid the interference of pentobarbital on MWM and TLR4/NF-κB, the animals that received pentobarbital were not used for behavioral or molecular analysis. The rat numbers (*n*) are included in each figure.

### 2.3. REM Sleep Deprivation

The modified multi-platform sleep-deprivation method was used to prepare the REM sleep-deprivation model. The REM sleep-deprivation rat boxes had a length of 120 cm, width of 60 cm and height of 40 cm. There were six platforms inside, and water (24–26 °C) was filled around the platform. Each platform was separated by 10 cm, and the rats could jump from one platform to another. The rats could eat, drink and move freely on the platform. The distance from the water surface to the platform was about 1 cm. When rats enter REM sleep, the muscle tone of the body decreases, which causes an imbalance, and the rats wake up as soon as they hit the water. The sleep-deprivation chamber ensured that the rats could not enter REM sleep. REM sleep deprivation was induced for 72 h [33]. The rats were subjected to behavioral testing after REM sleep deprivation.

### 2.4. Sodium Pentobarbital-Induced Sleeping

At 30 min, after the last time of oral administration of GAS and normal saline, five randomly selected rats in each group were intraperitoneally injected with sodium pentobarbital (35 mg/kg) dissolved in normal saline [34]. When the rats were placed in a supine position and could not be righted for more than 30 s, it was considered that the righting reflex disappeared. Sleep occurs when the righting reflex disappears, and awakening occurs when the righting reflex returns. Sleep latency was defined as the elapsed time between the administration of sodium pentobarbital and the disappearance of the righting reflex of the rats, and the time from the disappearance of the righting reflex to awakening was recorded as the sleep duration. The time was recorded with stopwatches.

### 2.5. Morris Water Maze

After the 72 h REM sleep deprivation, ten rats in each group were subjected to a Morris water maze experiment. The Morris water maze (MWM) was conducted as described previously [22]. A circular pool (diameter 120 cm, height 50 cm) was used and filled with water. The water temperature was maintained at 24–26 °C. One kilogram of milk powder was added to make the water opaque. A curtain was placed around the pool, and markers were placed on the curtain to provide clues to the rats, and these markers were not changed during the experiment. For the navigation test, a circular platform was placed 2 cm underwater in the third quadrant, and the position of the platform was fixed. The rats were placed in the water facing the pool wall at the entry points specified in quadrants 1, 2 and 4. The rats that found the target platform in 90 s were placed on the platform for 30 s. The rats that did not find the platform during the 90 s were artificially placed on the platform for 30 s to enhance their memory. The escape latency of the rats was recorded with 4 days as the experimental period. For the space exploration experiment, the circular platform was removed, and the rats were placed in water at the water entry point in quadrants 1, 2 and 3, and their residence time in the target quadrant (quadrant 4) during 90 s was recorded. The rats’ performance The locomotion trajectories of rats were recorded with a computer-based video tracking system and analyzed with MT-200 image analysis software (Taimeng Co., Ltd., Chengdu, China). At the end of each training, the rat hair was wiped clean, and the rats were placed in the cage to keep warm.

### 2.6. HE Staining

The rats were sacrificed after the behavior test, and the brain tissues were removed and immersed in 4% paraformaldehyde at 4 °C for more than 72 h. The portion of the brain tissue that included the whole hippocampus was embedded in paraffin. The paraffin-embedded brain tissue was then cut into 4–5 µm sections. Consequently, the sections were dewaxed with xylene, dehydrated with gradient alcohol, and finally stained with hematoxylin and erosion. Light microscopy was used to observe the morphological changes of the hippocampal neurons in the cerebral cortex and hippocampus CA1 region.

### 2.7. Western Blot Analysis

Hippocampal tissue was cleaved in a RIPA Lysis Buffer System that included protease inhibitors. After centrifugation (12,000 rpm, 20 min at 4 °C), the supernatant protein was measured using BCA protein assay reagent. Equal amounts of protein (30 μg) were separated with SDS–PAGE and transferred to PVDF membranes. The PVDF membranes were soaked in 5% skim milk for 1 h at room temperature and then incubated with primary anti-TLR4 antibody (1:1000 dilution), anti-p-p65 antibody (1:1000 dilution), anti-p65 antibody (1:1000 dilution), anti-p-IκBα antibody (1:1000 dilution), anti-IκBα antibody (1:1000 dilution), anti-Wnt3a antibody (1:1000 dilution) and anti-β-catenin antibody (1:1000 dilution) (Abcam, Cambridge, UK) at 4 °C overnight. After washing three times, the PVDF membranes were incubated with horseradish peroxidase-conjugated anti-rabbit or anti-mouse IgG secondary antibodies for 2 h at room temperature. Protein bands were detected using a chemiluminescence kit and visualized using Gel Imaging (Bio-Rad, Hercules, CA, USA).

### 2.8. Statistical Analysis

Repeated measures of multivariate analysis of variance were used to assess the results of MWM; others were analyzed using one-way analysis of variance with SPSS 20 (IBM, New York, USA) followed by a Tukey HSD multiple comparison test. A value of *p* < 0.05 is considered statistically significant.

## 3. Results

### 3.1. GAS Improved Sleep Latency and Sleep Duration in REM Sleep-Deprived Rats

Sleep latency and sleep duration are shown in Figure 1B,1C. Compared with the control group, the sleep latency was prolonged (*p* < 0.001) and the sleep duration (*p* < 0.001) was shortened in the model group. As expected, GAS was able to reduce pentobarbital-induced sleep latency (GAS 100 and 150: *Ps* < 0.001) and increase the duration of sleep (GAS 100: *p* = 0.011; GAS 150: *p* = 0.001).

### 3.2. GAS Prevents Cognitive Impairment in REM Sleep-Deprived Rats

The results from the Morris water maze test are shown in Figure 2A; the escape latency changed with training time (*p* < 0.001), and the effect of the group is also significant (*p* = 0.043). Compared with the control group, the duration of escape latency in the REM sleep-deprivation model group was longer (Figure 2A, day 1: *p* = 0.033; day 2: *p* = 0.001; day 3: *p* = 0.002; day 4: *p* < 0.001). Moreover, the time spent in the target quadrant of the REM sleep-deprivation model group was shorter than that of the control group (Figure 2B, *p* < 0.001). The treatment of the rats with GAS significantly prevented cognitive impairment, reduced the duration of escape latency to reach the hidden platform in day 2 (Figure 2A, GSA 150: day 2: *p* = 0.007; day 3 and day 4: *Ps* < 0.001) and increased the time spent in the target quadrant of the REM sleep-deprived rats (Figure 2B, GSA 100 and GSA 150: *Ps* < 0.001).

### 3.3. GAS Attenuates Pathological Damage in the Hippocampus and Cortex of REM Sleep-Deprived Rats

The hippocampal CA1 region and the cortex of the rats were observed with HE staining (Figure 3). Neuron loss, shrinkage, and severe cellular edema were observed both in the CA1 areas of the hippocampus and the cortex of the REM sleep-deprivation model group. However, treatment with GAS attenuated REM sleep-deprivation-induced neuronal damage, especially in the GAS (150 mg/kg) group.

### 3.4. GAS Suppressed the TLR4/NF-κB Signaling Pathway in REM Sleep-Deprived Rats

We next examined the effect of GAS on the TLR4/NF-κB signaling pathway. The levels of TLR4, p-p65, p65, p-IκBα and IκBα were determined with Western blot. The results indicate that the levels of TLR4, p-p65 and p-IκBα were strikingly improved in the sleep-deprived rats compared with those in the control rats (TLR4: *p* = 0.005; p-p65: *p* = 0.011; p-IκBα: *p* < 0.001), whereas GAS treatment dramatically reversed this effect (Figure 4) (GAS 100—TLR4: *p* = 0.014, p-p65: *p* = 0.038, p-IκBα: *p* < 0.001; GAS 150—TLR4: *p* = 0.001, p-p65 and p-IκBα: *Ps* < 0.001), indicating that the activation of the TLR4/NF-κB signaling pathway in the REM sleep-deprived rats was markedly inhibited by GAS treatment.

### 3.5. GAS Activated the Wnt/β-Catenin Signaling Pathway in REM Sleep-Deprived Rats

We next examined the effect of GAS on the Wnt/β-catenin signaling pathway. The levels of Wnt3a and β-catenin were determined using Western blot. The results indicate that the levels of Wnt3a and β-catenin were strikingly decreased in the REM sleep-deprived rats compared with those in the control rats (Wnt3a: *p* = 0.005; β-catenin: *p* = 0.007), whereas GAS treatment dramatically reversed this effect (Figure 5) (GAS 100—Wnt3a: *p* = 0.002, β-catenin: *p* = 0.040; GAS 150—Wnt3a: *p* < 0.001, β-catenin: *p* = 0.012), indicating that the activation of the Wnt/β-catenin signaling pathway in the REM sleep-deprived rats was markedly inhibited by GAS treatment.

## 4. Discussion

The present study reveals that GAS can reduce pentobarbital-induced sleep latency, increase the duration of sleep and effectively improve cognitive deficits and hippocampus CA1 and cortex neuron damage in REM sleep-deprived rats, and the mechanism is related to GAS inhibition of the TLR4/NF-κB signaling pathway and activation of the Wnt/β-catenin signaling pathway. This study is among the first to demonstrate the REM sleep-improvement effect of GAS involving the regulation of the TLR4/NF-κB and Wnt/β-catenin signaling pathways.

Sodium pentobarbital can selectively inhibit the central nervous system and have a sedative and hypnotic effect. A previous study has shown that GAS can significantly improve sleep in p-chlorophenylalanine-induced insomnia mice [19]. In our present study, we also found that GAS has a synergistic hypnotic effect on the threshold dose of sodium pentobarbital in REM sleep-deprived rats, and this synergistic hypnotic effect occurs in a dose-dependent manner, suggesting that GAS has a hypnotic effect by inhibiting the central nervous system.

Sleep deprivation, especially of REM sleep, can affect hippocampal synaptic plasticity and hippocampal long-term potentiation (LTP), contributing to a deficit in cognition [9,35,36]. Consistent with previous studies, in our present study, REM sleep-deprivation-exposed rats showed cognitive impairment in an MWM test. Fortunately, we found GAS can improve cognitive impairment caused by REM sleep deprivation.

Neuronal damage was found in the hippocampus CA1 region and the cerebral cortex in REM sleep-deprived rats [37,38]. In this study, pre-administration of GAS ameliorated neuronal damage in the hippocampus CA1 region and the cerebral cortex. The HE-staining results are consistent with the behavioral results of MWM. These results clearly demonstrate that GAS can improve sleep disturbance and cognitive impairment caused by REM sleep deprivation.

Neuroinflammation plays an important role in REM sleep-deprived cognitive impairment [39]. Previous studies have found that GAS has the effect of reducing neuroinflammation [40,41]. Therefore, we further explored whether GAS ameliorates REM sleep-deprivation-induced cognitive impairment by inhibiting neuroinflammation. The TLR4/NF-κB signaling pathway plays an important role in the transmission of neuroinflammatory signals. TLR4 is a protein that plays a natural role in immune recognition by recognizing the cell wall components of fungi and viruses. When TLR4 binds to the corresponding receptor, it can activate the downstream NF-κB signaling pathway [42]. The upregulation of NF-κB can induce the expression of proinflammatory cytokine genes and ultimately lead to neuronal injury. REM sleep deprivation activates the TLR4/NF-κB pathway, which is responsible for impaired learning and memory [43]. Consistent with previous studies, the expressions of TLR4, p-p65 and p-IκBα were increased in the model group. In addition, GAS significantly reduced the level of TLR4, p-p65 and p-IκBα. It is indicated that GAS ameliorated REM sleep-deprivation-induced cognitive impairment and neuronal injury by inhibiting the TLR4/NF-κB signaling pathway.

Wnt/β-catenin signaling also plays a key role in the cognitive dysfunction caused by REM sleep deprivation [44]. Wnt/β-catenin signaling contributes to neuroprotection, synapse formation and neurogenesis in the central nervous system [45]. The Wnt/β-catenin pathway comprises the extracellular signal, membrane segment, cytoplasmic segment and nuclear segment. Extracellular signals are mainly mediated by Wnt proteins, including Wnt3a. The cytoplasmic and the nuclear segment mainly include β-catenin, which translocates to the nucleus and activates downstream target genes [46]. Wnt/β-catenin-signaling-mediated hippocampal neuron genesis is involved in the cognitive decline caused by sleep deprivation. Therefore, the protein expression of Wnt3a and β-catenin was examined in this study. The Wnt/β-catenin signaling pathway is inhibited in REM sleep-deprived rats, and after pre-administration of GAS, Wnt/β-catenin signaling is activated. This indicates that GAS ameliorated REM sleep-deprivation-induced cognitive impairment and neuronal injury and is associated with activation of the Wnt/β-catenin pathway.

The major limitations of the present study are the lack of circadian rhythm and inflammatory cytokine measurements, which are the next goals for investigation.

## 5. Conclusions

In summary, the present study clearly demonstrates that GAS is beneficial in improving sleep quality and cognitive impairment and reducing neuronal damage and loss in sleep-deprived rats. These beneficial effects appear to be mediated, at least in part, by the regulation of the TLR4/NF-κB and Wnt/β-catenin signaling pathways.

## Figures and Tables

**Figure 1 brainsci-13-00179-f001:**
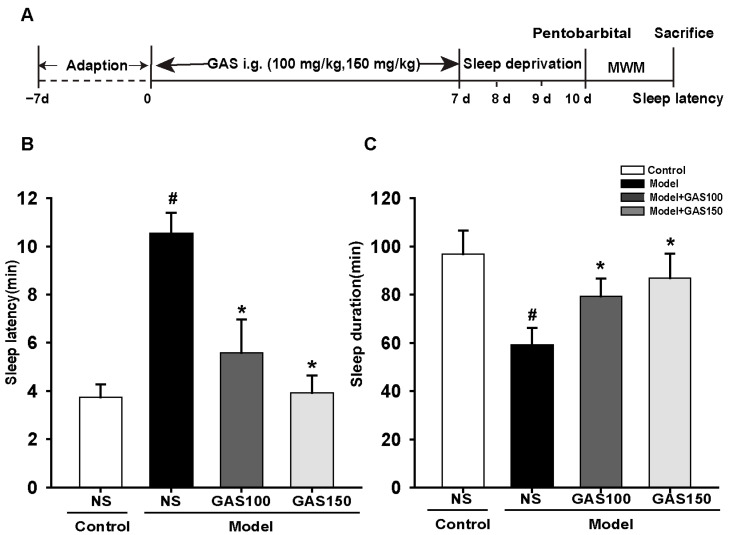
Effects of GAS on REM sleep-deprivation-induced sleep latency and sleep duration in rats: (**A**) the experiment design; (**B**) sleep latency; (**C**) sleep duration. Control + NS group: SD rats were lavaged with normal saline; Model + NS group: REM sleep-deprived rats were lavaged with normal saline; Model + GAS100, GSA150 groups: REM sleep-deprived rats were lavaged with gastrodin (100 or 150 mg/kg). ^#^
*p* < 0.05, versus control group; * *p* < 0.05, versus model group; (mean ± SD, *n* = 5).

**Figure 2 brainsci-13-00179-f002:**
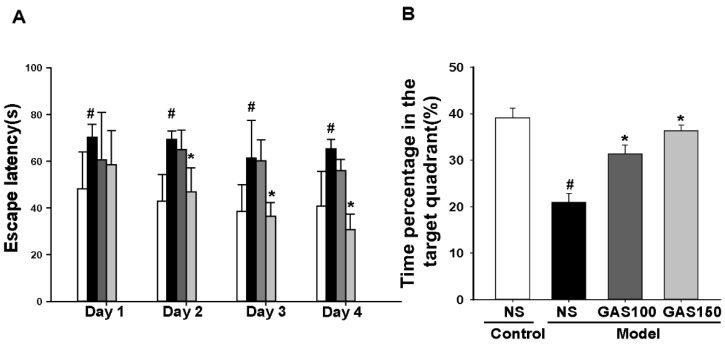
Effects of GAS on cognitive deficits in REM sleep-deprived rats: (**A**) the escape latency for 4 continuous days; (**B**) time spent in the target quadrant. ^#^
*p* < 0.05, versus control group; * *p* < 0.05, versus model group; (mean ± SD, *n* = 7–10).

**Figure 3 brainsci-13-00179-f003:**
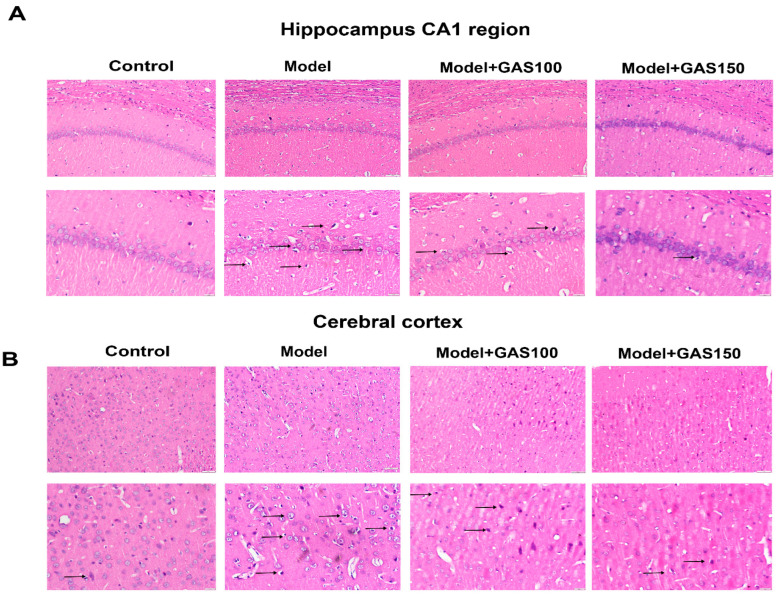
Effects of GAS on the pathological damage of neurons in the hippocampal CA1 region and cerebral cortex in REM sleep-deprived rats: (**A**) representative HE staining in hippocampus CA1 region; (**B**) representative HE staining in the cerebral cortex. The top row of pictures has a magnification of 400, bar = 50 μm; the bottom row of pictures has a magnification of 200, bar = 20 μm; *n* = 4.

**Figure 4 brainsci-13-00179-f004:**
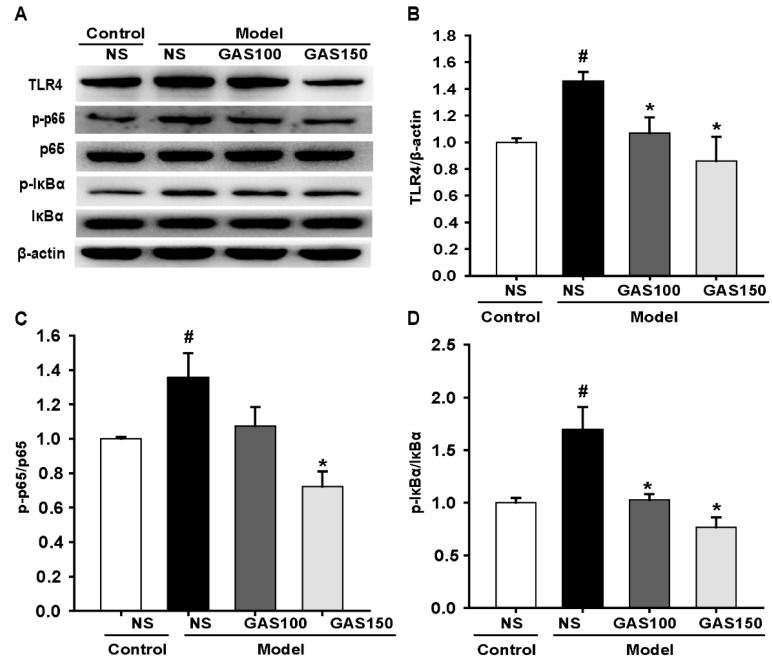
Effects of GAS on TLR4/NF-κB signaling pathway in REM sleep-deprived rats: (**A**,**B**) expression of TLR4, p-p65, p65, p-IκBα, IκBα proteins in the hippocampus; (**B**–**D**) quantitation of TLR4, p-p65 and p-IκBα levels. ^#^
*p* < 0.05, versus control group; * *p* < 0.05, versus model group; (mean ± SD, *n* = 3).

**Figure 5 brainsci-13-00179-f005:**
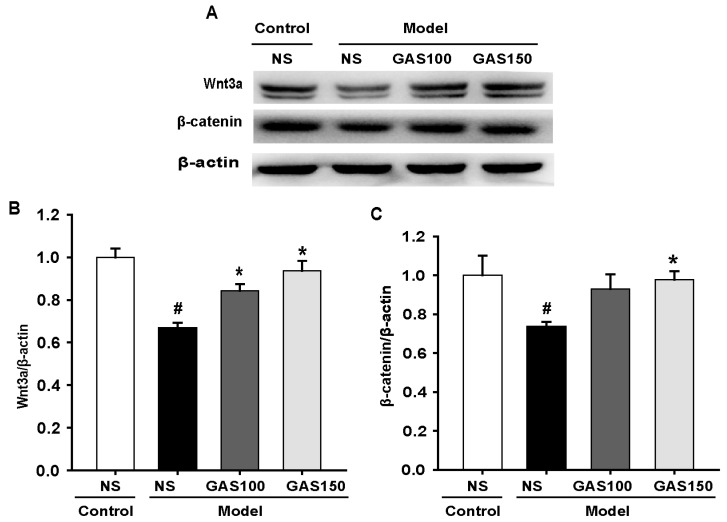
Effects of GAS on Wnt/β-catenin signaling pathway in REM sleep-deprived rats: (**A**) expression of Wnt3a and β-catenin proteins in the hippocampus; (**B**,C) quantitation of Wnt3a and β-catenin levels. ^#^
*p* < 0.05, versus control group; * *p* < 0.05, versus model group; (mean ± SD, *n* = 3).

## Data Availability

The data underlying this article are available in the article.

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
