# Peer review of "Gastrodin Improves Cognitive Dysfunction in REM Sleep-Deprived Rats by Regulating TLR4/NF-κB and Wnt/β-Catenin Signaling Pathways"

_brainsci, 2023, doi:10.3390/brainsci13020179_

Round 1
Reviewer 1 Report
Liu and colleagues present the results of a very interesting study in which they evaluated the effects of sleep deprivation on cognitive functions and evaluated involved mechnisms.
Page 2, lines 45-46 – it seems that a correction is needed as there is no medication such as Lorazepine. Did you mean lorazepam? If so, it belongs to benzodiazepines, so it does not need to be named separately. Similarly, estazolam is a benzodiazepine derivative, and naming it separately is misleading. Instead, you could name z-agents, such as zolpidem or SSRI, which improve sleep, such as trazodone, agomelatine, or mirtazapine. Additionally, a statement following this needs to be rephrased, and references need to be provided, as available data clearly states that while benzodiazepines help with the induction of sleep, they decrease sleep quality and may cause cognitive dysfunction [lines 46-47].
Other statements in the introduction also need a reference.
An expansion of the description of molecular pathways – and why they were chosen as well as a mechanism involved in immunological complications following sleep deprivation would be advantageous for the manuscript (f.e. doi:10.3390/biomedicines10092159 and doi: 10.3390/jcm11010067).
The materials and method section needs a description of how the sleep data was recorded and how it was scored – this is crucial information.
In lines 104-106, you state that 40 rats were divided into 4 study groups, and each study group consisted of 15 subjects. This does not add up. Please correct it.
In the description of sleep deprivation, you state that REM sleep could not be achieved by the rats. Was the deprivation only limited to REM sleep, or was it for all stages of sleep – this needs to be cleared out as it is crucial for the study. If it was limited only to REM sleep, the manuscript needs to be corrected in this context, as REM sleep deprivation is not the same as total sleep deprivation.
While citing SPSS in brackets, provide the company (IBM) instead of repeating the name of the program.
Please and a figure describing each of the study groups, what was administered and what procedure was undertaken, as it is hard to follow while reading the results which group is which. What does the model group consist of?
The title of figure 2 needs to be changed as the present title is a description of the figure, not a title.
Expand all abbreviations, f.e SPF (line 98).
A general correction of the English throughout the manuscript is needed as well as correction of several misspellings.
Author Response
[Comment] 1. Page 2, lines 45-46 it seems that a correction is needed as there is no medication such as Lorazepine. Did you mean lorazepam? If so, it belongs to benzodiazepines, so it does not need to be named separately. Similarly, estazolam is a benzodiazepine derivative, and naming it separately is misleading. Instead, you could name z-agents, such as zolpidem or SSRI, which improve sleep, such as trazodone, agomelatine, or mirtazapine. Additionally, a statement following this needs to be rephrased, and references need to be provided, as available data clearly states that while benzodiazepines help with the induction of sleep, they decrease sleep quality and may cause cognitive dysfunction [lines 46-47].
Response: Thanks for the comment. In the revised manuscript, we have changed Page 2, lines 45-46 as: “At present, sedative and hypnotic drugs, such as benzodiazepines [11], zolpidem [12], and selective serotonin reuptake inhibitor trazodone, agomelatine and mirtazapine [13,14], are widely used to improve sleep problems.” Four new references haven been added: de Mendonca 2021 [11]; Simon 2022 [12]; Wichniak 2011 [13]; de Oliveira 2022 [14].
[Comment] 2. The materials and method section needs a description of how the sleep data was recorded and how it was scored – this is crucial information.
Response: Thanks for the comment. We have added a description as “Mice were placed in supine position and could not be righted for more than 30 s, it was considered that the righting reflex disappeared. Sleep occurs when the righting reflex disappears and awakening occurs when the righting reflex returns. Sleep latency was defined as the elapsed time between the administration of sodium pentobarbital and the disappearance of righting reflex of rats, and the time from the disappearance of righting reflex to the awakening was recorded as the sleep duration. The time is recorded with a stopwatch.” in the materials and method section 2.4.
[Comment] 3. In lines 104-106, you state that 40 rats were divided into 4 study groups, and each study group consisted of 15 subjects. This does not add up. Please correct it.
Response: Thanks for the comment. We have corrected “Forty Sprague Dawley rats” as “Sixty Sprague Dawley rats”.
[Comment] 4. In the description of sleep deprivation, you state that REM sleep could not be achieved by the rats. Was the deprivation only limited to REM sleep, or was it for all stages of sleep – this needs to be cleared out as it is crucial for the study. If it was limited only to REM sleep, the manuscript needs to be corrected in this context, as REM sleep deprivation is not the same as total sleep deprivation.
Response: Thanks for the comment. “Sleep deprivation” have been corrected as“REM sleep deprivation” in the context and in the Methods as “When the rats enter REM sleep, the muscle tone of the body decreases, which causes an imbalance and the rats wake up as soon as they hit water. The sleep deprivation chamber ensured that the rats could not enter REM sleep. REM sleep deprivation was induced for 72 h[33].”
[Comment] 5. While citing SPSS in brackets, provide the company (IBM) instead of repeating the name of the program.
Response: Thanks for the comment. “SPSS 20 (SPSS, United States)” have been corrected as “SPSS 20 (IBM, USA)”.
[Comment] 6. Please and a figure describing each of the study groups, what was administered and what procedure was undertaken, as it is hard to follow while reading the results which group is which. What does the model group consist of ?
Response: Thanks for the comment. “Control+NS group: SD rats were lavaged with normal saline; Model+NS group: REM sleep deprivation rats were lavaged with normal saline; Model+GAS100, GSA150 groups: REM sleep deprivation rats were lavaged with gastrodin (100 or 150 mg/kg).” have been added to the Figure 1 legend.
[Comment] 7. The title of figure 2 needs to be changed as the present title is a description of the figure, not a title.
Response: Thanks for the comment. The title of figure 2 have been changed as“Effects of GAS on cognitive deficits in REM sleep deprived rats”
[Comment] 8. Expand all abbreviations, f.e SPF (line 98).
Response: Thanks for the comment. Corrected.
[Comment] 9. A general correction of the English throughout the manuscript is needed as well as correction of several misspellings.
Response: Thanks for the comment. We have made careful proofreading with English professionals.

Reviewer 2 Report
“Gastrodin improves cognitive dysfunction in sleep-deprived rats by regulating TLR4/NF-κB and Wnt/β-catenin signaling pathways”(brainsci-2136751)
This manuscript aimed to explore the effects of Gastrodin on cognitive function in sleep-deprived rats and the potential TLR4/NF-κB and Wnt/β-catenin signaling pathways. The results revealed that Gastrodin treatment significantly improved sleep deprivation-induced sleep disturbance, cognitive deficits and neuron damage in hippocampus CA1 region and cerebral cortex. Mechanism analysis revealed that NF-κB pathway was activated and Wnt/β-catenin pathway was inhibited after sleep deprivation, and Gastrodin ameliorated these aberrant changes. Overall, this topic is interesting and the results are promising. However, some concerns appeared after reading the whole manuscript.
1. The current investigation is not involved human, thus, the literature review should be more focused on animal studies. The following references might be helpful.
Hu, B., Liu, C., Lv, T., Luo, F., Qian, C., Zhang, J., ... & Liu, Z. (2022). Meta-analysis of sleep deprivation effects on depression in rodents. Brain Research, 1782, 147841.
Pires, G. N., Bezerra, A. G., Tufik, S., & Andersen, M. L. (2016). Effects of experimental sleep deprivation on anxiety-like behavior in animal research: Systematic review and meta-analysis. Neuroscience & Biobehavioral Reviews, 68, 575-589.
Küpeli Akkol, E., Bardakcı, H., Yücel, Ç., Åžeker Karatoprak, G., Karpuz, B., & Khan, H. (2022). A New Perspective on the Treatment of Alzheimer’s Disease and Sleep Deprivation-Related Consequences: Can Curcumin Help?. Oxidative Medicine and Cellular Longevity, 2022.
Wu, H., Dunnett, S., Ho, Y. S., & Chang, R. C. C. (2019). The role of sleep deprivation and circadian rhythm disruption as risk factors of Alzheimer’s disease. Frontiers in neuroendocrinology, 54, 100764.
Agrawal, S., Kumar, V., Singh, V., Singh, C., & Singh, A. (2023). A review on pathophysiological aspects of sleep deprivation. CNS & Neurological Disorders-Drug Targets (Formerly Current Drug Targets-CNS & Neurological Disorders).
Milman, N. E., Tinsley, C. E., Raju, R. M., & Lim, M. M. (2022). Loss of sleep when it is needed most–Consequences of persistent developmental sleep disruption: A scoping review of rodent models. Neurobiology of Sleep and Circadian Rhythms, 100085.
Zamore, Z., & Veasey, S. C. (2022). Neural consequences of chronic sleep disruption. Trends in Neurosciences.
Villafuerte, G., Miguel-Puga, A., Murillo Rodríguez, E., Machado, S., Manjarrez, E., & Arias-Carrión, O. (2015). Sleep deprivation and oxidative stress in animal models: a systematic review. Oxidative medicine and cellular longevity, 2015.
2. some related references were missing and should be reviewed and discussed in the current manuscript, such as,
Arvin, P., Ghafouri, S., Bavarsad, K., Hajipour, S., Khoshnam, S. E., Sarkaki, A., & Farbood, Y. (2023). Therapeutic effects of growth hormone in a rat model of total sleep deprivation: Evaluating behavioral, hormonal, biochemical and electrophysiological parameters. Behavioural Brain Research, 438, 114190.
Tang, H., Li, K., Dou, X., Zhao, Y., Huang, C., & Shu, F. (2020). The neuroprotective effect of osthole against chronic sleep deprivation (CSD)-induced memory impairment in rats. Life Sciences, 263, 118524.
Arora, S., Dharavath, R. N., Bansal, Y., Bishnoi, M., Kondepudi, K. K., & Chopra, K. (2021). Neurobehavioral alterations in a mouse model of chronic partial sleep deprivation. Metabolic Brain Disease, 36(6), 1315-1330.
Liu, Y., Gao, J., Peng, M., Meng, H., Ma, H., Cai, P., ... & Si, G. (2018). A review on central nervous system effects of gastrodin. Frontiers in Pharmacology, 9, 24.
Ye, T., Meng, X., Wang, R., Zhang, C., He, S., Sun, G., & Sun, X. (2018). Gastrodin alleviates cognitive dysfunction and depressive-like behaviors by inhibiting ER stress and NLRP3 inflammasome activation in db/db mice. International Journal of Molecular Sciences, 19(12), 3977.
Chen, T. T., Zhou, X., Xu, Y. N., Li, Y., Wu, X. Y., Xiang, Q., ... & Shen, X. C. (2021). Gastrodin ameliorates learning and memory impairment in rats with vascular dementia by promoting autophagy flux via inhibition of the Ca2+/CaMKII signal pathway. Aging (Albany NY), 13(7), 9542.
Deng, C., Chen, H., Meng, Z., & Meng, S. (2022). Gastrodin and Vascular Dementia: Advances and Current Perspectives. Evidence-Based Complementary and Alternative Medicine, 2022.
Wang, X., Li, S., Ma, J., Wang, C., Chen, A., Xin, Z., & Zhang, J. (2019). Effect of gastrodin on early brain injury and neurological outcome after subarachnoid hemorrhage in rats. Neuroscience bulletin, 35(3), 461-470.
Wang, S., Nan, Y., Zhu, W., Yang, T., Tong, Y., & Fan, Y. (2018). Gastrodin improves the neurological score in MCAO rats by inhibiting inflammation and apoptosis, promoting revascularization. International journal of clinical and experimental pathology, 11(11), 5343.
Li, Y., Zhang, E., Yang, H., Chen, Y., Tao, L., Xu, Y., ... & Shen, X. (2022). Gastrodin Ameliorates Cognitive Dysfunction in Vascular Dementia Rats by Suppressing Ferroptosis via the Regulation of the Nrf2/Keap1-GPx4 Signaling Pathway. Molecules, 27(19), 6311.
WU, D. N., DING, R. C., JI, K., WANG, P., & LIU, L. (2020). Effect of Suanzaoren Tang on learning memory and TLR4/NF-κB signaling pathway in rats with chronic sleep deprivation. Chinese Journal of Experimental Traditional Medical Formulae, 18-24
Luo, Y., Chen, P., Yang, L., & Duan, X. (2022). Network pharmacology and molecular docking analysis on molecular targets and mechanisms of Gastrodia elata Blume in the treatment of ischemic stroke. Experimental and Therapeutic Medicine, 24(6), 1-16.
Wang, D., & Dong, X. (2021). Gastrodin provides neuroprotection in models of traumatic brain injury via Nrf2 signaling pathway. Quality Assurance and Safety of Crops & Foods, 13(4), 62-69.
Nabaee, E., Kesmati, M., Shahriari, A., Khajehpour, L., & Torabi, M. (2018). Cognitive and hippocampus biochemical changes following sleep deprivation in the adult male rat. Biomedicine & Pharmacotherapy, 104, 69-76.
3. The novelty of the current investigation should be stated more clearly.
4. How did you determine the sample size? Did you calculate the sample size needed before formal study?Although you mentioned that the current sample size was based on your prior publications [24; 25], however, the sample sizes used in your prior publications were not well justified.
References:
Lakens, D. (2022). Sample size justification. Collabra: Psychology, 8(1), 33267.
5. Please provide the descriptive data and the effect sizes where available in the formal context. For the p value, please provide the specific data unless the p < 0.001.
6. “The present study revealed that GAS could effectively improve the sleep disorder” this statement is not appropriate.
7. “Previous studies have shown that GAS can significantly improve sleep in p-chlorophenylalanine-induced insomnia mice[14].” You mentioned “studies”, however, you only cite one study. This is also the case for “Our previous studies have found that GAS has the effect of reducing neuroinflammation[15].”
8. Limitation part should be added.
9. Line 98: what does “SPF” mean?
Author Response
This manuscript aimed to explore the effects of Gastrodin on cognitive function in sleep-deprived rats and the potential TLR4/NF-κB and Wnt/β-catenin signaling pathways. The results revealed that Gastrodin treatment significantly improved sleep deprivation-induced sleep disturbance, cognitive deficits and neuron damage in hippocampus CA1 region and cerebral cortex. Mechanism analysis revealed that NF-κB pathway was activated and Wnt/β-catenin pathway was inhibited after sleep deprivation, and Gastrodin ameliorated these aberrant changes. Overall, this topic is interesting and the results are promising. However, some concerns appeared after reading the whole manuscript.
Response: Thanks for positive remarks.
[Comment] 1. The current investigation is not involved human, thus, the literature review should be more focused on animal studies. The following references might be helpful.
Response: Thanks for the comment. We appreciate the suggested references. Most of the above literature has been cited in the revised manuscript: Hu 2022 [7]; Pries 2012 [2]; Kupeli Akkol 2022 [4]; Wu, 2019 [5]; Agrawal 2022 [11]; Milman, 2023 [10]; Zamore, 2022 [35]; and Villafuerte, 2015 [36].
[Comment] 2. Some related references were missing and should be reviewed and discussed in the current manuscript, such as,
Response: Thanks for the comment. We appreciate the suggested references. Most of the above literature has been cited in the revised manuscript: Liu, 2018 [18]; Ye, 2018 [40]; Deng, 2022 [23]; Wang, 2018 [41]; Li, 2022 [24]; and Luo, 2022 [21]. In the revised manuscript, we have increased references from 35 to 46, with 29 new added references and removed 17 old references to make the manuscript more updated.
[Comment] 3. The novelty of the current investigation should be stated more clearly.
Response: In the introduction of the revised manuscript, we have added “Our research group and others researchers have demonstrated that GAS is able to ameliorate cerebral ischemia-reperfusion injury [20,21] and chronic cerebral hypoperfusion injury [22-24]. Nevertheless, the effect of GAS on learning and memory impairment caused by REM sleep deprivation has not been studied, and is the main goal of the present study. We hypothesized that GAS could improve learning and memory impairment in rats with REM sleep deprivation, and the mechanism might be related to the regulation of TLR4/NF-κB and Wnt/β-catenin signaling pathways.
[Comment] 4. How did you determine the sample size? Did you calculate the sample size needed before formal study?Although you mentioned that the current sample size was based on your prior publications [24; 25], however, the sample sizes used in your prior publications were not well justified.
Response: Thanks for the comment. The dose section is based on our prior publication [22] and the literature of GAS to affect TLR4/NF-κB [32]. To avoid the interference of pentobarbital on MWM and TLR4/NF-κB, animals received pentobarbital were not used for behavioral and molecular analysis. The rat numbers (n) were included in each Figure Legend.
[Comment] 5. Please provide the descriptive data and the effect sizes where available in the formal context. For the p value, please provide the specific data unless the p < 0.001.
Response: The descriptive data and the effect sizes have been provided in the revised manuscript.
[Comment] 6. “The present study revealed that GAS could effectively improve the sleep disorder” this statement is not appropriate.
Response: “effectively improve the sleep disorder” have been corrected as “could reduce pentobarbital-induced sleep latency and increase the duration of sleep”.
[Comment] 7. “Previous studies have shown that GAS can significantly improve sleep in p-chlorophenylalanine-induced insomnia mice[14].” You mentioned “studies”, however, you only cite one study. This is also the case for “Our previous studies have found that GAS has the effect of reducing neuroinflammation[15].”
Response: Thanks for the comment. Corrected as “Previous study has shown that GAS can significantly improve sleep in p-chlorophenylalanine-induced insomnia mice[19],” and “Previous studies have found that GAS has the effect of reducing neuroinflammation[40,41].”
[Comment] 8. Limitation part should be added.
Response: “The major limitations of the present study are the lack of circadian rhythm and inflammatory cytokine measurements which are next goals of investigation” has been added at the end of Discussion.
[Comment] 9. Line 98: what does “SPF” mean?
Response: Corrected.

Round 2
Reviewer 2 Report
Thanks for the revisions and no further concerns.